# First comprehensive analysis of *Aedes aegypti* bionomics during an arbovirus outbreak in west Africa: Dengue in Ouagadougou, Burkina Faso, 2016–2017

**Athanase Badolo**[1]*, **Aboubacar Sombié**[1], **Félix Yaméogo**[1‡], **Dimitri W. Wangrawa**[1,2‡], **Aboubakar Sanon**[1], **Patricia M. Pignatelli**[3], **Antoine Sanon**[1], **Mafalda Viana**[4], **Hirotaka Kanuka**[5,6], **David Weetman**[3⊚], **Philip J. McCall**[3⊚]*

**1** Laboratoire d'Entomologie Fondamentale et Appliquée, Université Joseph Ki-Zerbo, Ouagadougou, Burkina Faso, **2** Université Norbert Zongo, Koudougou, Burkina Faso, **3** Department of Vector Biology, Liverpool School of Tropical Medicine, Liverpool, United Kingdom, **4** Institute of Biodiversity, Animal Health and Comparative Medicine, University of Glasgow, Glasgow, United Kingdom, **5** Department of Tropical Medicine, The Jikei University School of Medicine, Tokyo, Japan, **6** Center for Medical Entomology, The Jikei University School of Medicine, Tokyo, Japan

⊚ These authors contributed equally to this work.
‡ FY and DWW also contributed equally to this work.
* a.badolo@gmail.com (AB); philip.mccall@lstmed.ac.uk (PJM)

## Abstract

### Background

Dengue's emergence in West Africa was typified by the Burkina Faso outbreaks in 2016 and 2017, the nation's largest to date. In both years, we undertook three-month surveys of *Aedes* populations in or near the capital city Ouagadougou, where the outbreaks were centered.

### Methodology

In 1200LG (urban), Tabtenga (peri-urban) and Goundry (rural) localities, we collected indoor and outdoor resting mosquito adults, characterized larval habitats and containers producing pupae and reared immature stages to adulthood in the laboratory for identification. All mosquito adults were identified morphologically. Host species (from which bloodmeals were taken) were identified by PCR. Generalized mixed models were used to investigate relationships between adult or larval densities and multiple explanatory variables.

### Results

From samples in 1,780 houses, adult *Ae. aegypti* were significantly more abundant in the two urban localities (Tabtenga and 1200 LG) in both years than in the rural site (Goundry), where *Anopheles* spp. were far more common. Results from adult collections indicated a highly exophilic and anthropophilic (>90% bloodmeals of human origin) vector population, but with a relatively high proportion of bloodfed females caught inside houses. Habitats

**Data Availability Statement:** All relevant data are within the manuscript and its Supporting Information files.

**Funding:** The study was funded under a WHO/TDR fellowship awarded to AB in collaboration with DW and PJM under the reference number WHO/TDR/ RCS-KM 2015 ID235974. PJM's research on peri-domestic behavior of Aedes aegypti receives support from MRC-UK (MR/T001267/1). M.V. was funded by the European Research Council under the European Union's Horizon 2020 Research and Innovation Programme (grant agreement no. 852957). The funders had no role in study design, data collection and analysis, decision to publish, or preparation of the manuscript.

**Competing interests:** The authors have declared that no competing interests exist.

producing most pupae were waste tires (37% of total pupae), animal troughs (44%) and large water barrels (30%).

While Stegomyia indices were not reliable indicators of adult mosquito abundance, shared influences on adult and immature stage densities included rainfall and container water level, collection month and container type/purpose. Spatial analysis showed autocorrelation of densities, with a partial overlap in adult and immature stage hotspots.

## Conclusion

Results provide an evidence base for the selection of appropriate vector control methods to minimize the risk, frequency and magnitude of future outbreaks in Ouagadougou. An integrated strategy combining community-driven practices, waste disposal and insecticide-based interventions is proposed. The prospects for developing a regional approach to arbovirus control in West Africa or across Africa are discussed.

## Author summary

*Aedes aegypti* is the most efficient vector of human diseases including yellow fever, dengue chikungunya and Zika. West Africa is an emerging hotspot for dengue, as illustrated by Burkina Faso's outbreaks in 2016 and 2017. Coincidentally, this study investigated the bionomics and behavior of *Ae. aegypti* in urban, peri-urban and rural localities of Ouagadougou, from August to October in 2016 and 2017. The results from over 1700 homes showed that *Ae. aegypti* preferentially fed on humans and rested primarily outdoors. The most common pupal productive habitats were discarded vehicle tires, animal water troughs and large water barrels. Infestation rates were higher in the urban and peri-urban localities. A range of community-based control measures are suitable for consideration in a vector control program to prevent outbreaks.

This study provides the most complete contemporary description of an *Ae. aegypti* population in West Africa, providing evidence for mosquito control programs to prevent or mitigate disease outbreaks. The potential of a regional vector control program for west Africa is discussed briefly.

## Background

*Aedes aegypti* mosquitoes thrive in urban environments throughout the tropics and sub-tropics, where they are the primary human vectors of dengue, Zika, chikungunya and urban yellow fever arboviruses. Dengue is the most widespread with an estimated annual toll of 400 million infections, and a threat to almost half of the human population at risk globally [1,2]. Driven by climate change, further increases are predicted [3] and the search for effective and sustainable methods to control peridomestic *Aedes* populations has become a primary global health challenge.

The worldwide expansion in *Aedes aegypti* populations in the late twentieth century followed a period when the human population grew at an unprecedented rate, mainly in cities, environments where this vector also could flourish [4,5]. Increases in global trade and human travel, particularly access to affordable and rapid intercontinental air travel, transferred multiple strains of arboviruses far beyond their original ranges into immunologically naïve

populations [6], while climate change increased outbreak risk in new areas outside of the tropics [7,8]. With almost 500 million people already living in its cities and a population doubling-rate of 25 years, sub-Saharan Africa has the fastest urbanization growth rate worldwide [9,10]. In many countries, sylvatic cycles of yellow fever frequently initiate outbreaks [11–16] that threaten urban areas where human to human transmission by *Ae. aegypti* would result in frequent outbreaks, were it not for an effective vaccine [17,18]. There have been significant outbreaks of chikungunya in countries of East, South and North Africa [19–22] and increasingly, reports of Zika [23,24]. Indeed, better diagnostic methods are revealing that chikungunya infections, though causing little mortality, are responsible for greater morbidity and a higher-disease burden than previously suspected [6]. Dengue presents the greatest arboviral threat to Africa, with West Africa identified as a potential dengue transmission hotspot, based on its rapid rate of unplanned urbanization, widespread occurrence of *Aedes* vectors, history of arbovirus transmission and poor clinical knowledge of flavivirus infections [25].

The vast majority of recent knowledge of *Aedes aegypti* was generated by studies in Asia and Latin America [26–29], and may not be transferred reliably to African *Ae. aegypti* populations [30]. The need for continent-specific knowledge is reinforced by growing evidence of the genetic and phenotypic diversity of *Ae. aegypti* within Africa and differences in the epidemiology of *Aedes*-borne diseases globally [4,31,32].

Burkina Faso experienced dengue outbreaks in 2016, resulting in 2,600 cases and 21 deaths, and in 2017, resulting in 14,455 cases and 29 deaths [33,34]. The outbreak continued into 2018, with a further 4,386 cases and 25 deaths. Though all four dengue virus serotypes had been recorded previously in Burkina Faso [35], only types I, II and III were reported in the recent outbreak, of which type II was the most prevalent [36].

In August 2016, we began a study on the bionomics of *Ae. aegypti* in Ouagadougou, the capital city of Burkina Faso, with the objective of collecting baseline data for dengue control. Though not planned that way, the study coincided with the dengue outbreaks in 2016 and 2017 and sampling was carried out in locations where and while transmission was occurring. Here we report the results of wet season investigations into the behavioral preferences and spatial distribution of *Ae. aegypti* breeding and adult resting sites across different levels of urbanization, and use this evidence base to develop an appropriate, effective and sustainable vector control strategy for Ouagadougou, suitable for use as a regional template within a larger integrated vector management program.

## Materials and methods

### Ethics statement

The study protocol received ethical approval from the National Ethical Committee (Comité National d'Ethique pour la Recherche en Santé), Ministry of Health, Burkina Faso (Deliberation No. 2016-6-073; 6[th] June 2016) and from the Liverpool School of Tropical Medicine Research Ethics Committee for "Dengue in Burkina Faso: establishing a vector biology evidence base for risk assessment and vector control strategies for an emerging disease"(Research Protocol 16–30; 15[th] July 2016).

### Study area

Burkina Faso occupies three climatic zones: the Sahelian in the north, the central Soudano-Sahelian zone and Soudanian in the south. The capital city Ouagadougou lies in the Soudano-Sahelian zone, within the 500 mm isohyet where 350–750 mm rain falls between June and October. Three distinct localities in or near the city were selected on the basis of house design

and size, house density (peri-urban had higher housing density compared to urban and rural), and land use, as representatives of urban, peri-urban and rural settings.

## 1200 Logements

(1200 LG) (12˚22'N; 1˚29'W) is an urban setting of 1.2 km$^2$ in central Ouagadougou, less than 1 km from the international airport. Roads are paved and the area is connected to centralized water, waste and electricity systems. Houses are relatively modern single- or two-storey and typically comprise one living room and 2–3 bedrooms, often with air conditioning. Vegetation, often within gardens, is common on both private and common land.

## Tabtenga

(12˚22'N; 1˚27'W) is a 10 km$^2$ peri-urban setting located within Ouagadougou, approximately 5 km east of 1200 LG. Roads are unpaved, there are no electricity or waste management systems and the majority of households obtain water at communal pumps. Typical households are single-storey structures with 1–4 rooms, within walled compounds. Vegetation is sparse.

## Goundry

(12˚30'N, 1˚20'W) is a small rural farming community village situated 30 km north-east of Ouagadougou, with unpaved roads, low housing density, and is surrounded by fields and trees. There is a dam in the center of the villages enabling people to practice gardening during the dry season. Goundry has no electricity or waste management systems and households obtain water at communal pumps. Livestock, mainly cattle and sheep, and dogs are common.

## Study design

Longitudinal surveys were carried out during the wet season from August to October in both 2016 and 2017. Prior to each survey, all prospective houses were visited to inform the population of the proposed project, and the proposed sampling procedures. Each day, the first house was selected at random and the second and subsequent houses chosen where the family were present and agreed to participate, avoiding the houses nearest to the previous one. An average of ten houses were visited per day. Houses were sampled at 06:00–09:00 or 16:00–19:00 by two teams of four persons, each working as follows: one person collected written informed consent and household data (S1 Table) from participants, a second person searched for adult mosquitoes indoors and outdoors; two people recorded breeding habitat characteristics and collected immature stages of mosquitoes. Areas of public or communal land adjacent to sampled houses were mapped and inspected for containers, from which mosquito immature stages were processed as described below.

## Breeding site characterization and collection of immature mosquitoes

At each property, the team worked indoors and outdoors inspecting every container capable of holding sufficient water for immature mosquitoes, recording its dimensions, water volume, water level, material (natural, wood, metal, cement, etc.) and utility (whether the container was in use (yes) or discarded waste (no)). Where possible, water from each container was poured into a graduated beaker and any immature mosquitoes collected using a sieve. The water volume of heavy or immovable containers, including potable water, was measured by removal with buckets, again using a sieve to collect all mosquitoes, before being returning to the container. All larvae and pupae were transferred alive to containers labelled by house number and breeding site location, for subsequent identification.

Breeding habitats were categorized using the WHO operational guide [28] with minor adaptations to accommodate certain features observed in Ouagadougou as follows:

- Large containers—drums and barrels; water volume > 50 L

- Medium containers—buckets, large pots and small drums; 10–49 L

- Small containers—all container types of any material < 10 L,

- Tires—stored or discarded vehicle tires

- Drinking troughs—water for livestock, of any material

- Other–ground water puddle, tree holes and flower pots

   *Stegomyia* indices were calculated as defined by Focks[37]:

- House index (HI): defined as the percentage of houses infested with *Aedes aegypti* larvae and/or pupae in a locality

- Container index (CI): the percentage of water-holding containers positive for immature stages of *Aedes aegypti*

- Breteau index: the number of containers positive for *Aedes aegypti*/ 100 houses inspected

## Pupal indices

To identify the most productive dengue vector breeding sites, and to determine whether certain habitats that were relatively more productive for pupae might be identified for future targeting, we recorded pupae separately from larvae. Pupal mortality is typically low meaning that the number of pupae is highly correlated with the number of adults [38]. To identify the most productive *Ae. aegypti* immature stage habitats, the percentage contribution of each container type to the total count of pupae is calculated as the total number of pupae per container type, divided by the total number of pupae in all containers throughout the study area [28].

## Adult mosquito collection

Using a Prokopack aspirator [39], all adult mosquitoes were collected from each house, animal shelter and external kitchen. Indoors, wall surfaces, closet or cupboard interiors and other known resting places [40], were inspected using flashlights, for a total of 10 minutes per household [41]. Outdoors, all walls, eaves, vegetation (flowers) and shaded areas within or behind containers, stored materials, and car tires within the walled area marking the perimeter of each household were then inspected for an additional 10 minutes.

## Processing and identification of collected mosquitoes

All immature mosquito stages were sorted based on morphology then reared in cups with ground Tetramin as food until adult emergence. Pupae were transferred to fresh cups and emerging adults killed and preserved by freezing for identification.

   Mosquitoes were identified by microscopy using morphological keys. *Aedes aegypti* [42–44] was identified as described by Moore et al. [45] and all other culicine species were identified using Edwards (1941)[46]. *Anopheles* species identification followed the keys of Gillies & Coetzee [47] and Diagne et al. [48]. The SINE method [49] was used to identify *Anopheles gambiae* complex mosquitoes to species level.

## Identification of bloodmeal origin

DNA was extracted from abdomens of bloodfed females using Qiagen DNEasy kits. Extracted DNA served as template for amplification of the cytochrome b gene using human, cow, pig, dog, goat and sheep species-specific primers in a cocktail PCR [50]. Each reaction contained 15.8 ul of water, 2.5ul of 10x DreamTaq buffer, 0.5 ul 10mM of each dNTP, 0.5ul of 50 µM of each primer; 0.2 of 5U/ul DreamTaq and 3 ul of DNA for a total volume of 25ul. The PCRs were run with the program 95˚C for 5 min, then 50 cycles of 95˚C for 60 s, 56˚C for 60 s, 72˚C for 60 s, followed by a final extension step of 72˚C for 7 min. Products were run on a 2% TAE-agarose gel, with band size interpretation as follows: 334 bp (human), 453 bp (pig), 132 bp (goat or sheep), 680 bp (dog), 561 bp (cow). PCR products of positive bloodmeals that gave unclear gel banding-patterns were sequenced using universal vertebrate primers [50], aligned using codon code aligner v 4.7 (Codoncode corp., USA) and their species of origin identified using NCBI BLAST searches.

## Meteorological data

Daily records of minimal, maximal, mean temperatures, relative humidity and daily rainfall were obtained from the National Meteorological Agency records from Ouagadougou station for 2016 and 2017. Intermediate calculations were made for the cumulative rainfall for previous 4 days, one week, 12 days or 14 days to be included in the models.

## Statistical analyses

To investigate factors associated with *Aedes sp*. abundance we developed four Generalised Linear Mixed Models (GLMMs) with a negative binomial link function using the R package "glmmTMB" [51]. The first (adult model) and second (bloodfed model) were fitted to the number of adults and bloodfed *Aedes* collected in each house, respectively, as a function of the abiotic and biotic covariates: the year of collection, locality (1200 LG, Tabtenga, Goundry), house type (mixed, modern or traditional), location (indoors/outdoors), number of children, ITNs presence, ITNs number, animal presence, animal number, day of collection, month of collection, the number of immatures collected in breeding sites located in that house, and also climate factors including temperature and cumulative rainfall. The interaction terms 'collection location' and locality, and locality and year were also included, and to account for variation arising from the sampling design we included date of collection and house identifier as random effects. The third (larval model) and fourth (pupal model) models were fitted to the number of larvae and pupae per container, respectively, as a function of the abiotic and biotic covariates: locality, container type, water level, water volume, container material, the 'location' (i.e. indoors/outdoors) of the containers utility, rainfall and temperature, number of adults collected in the house. As in the adult mosquito model, the variables location and locality, and year and locality were also included as interaction terms, and date of collection and house identifier were added as random effects.

From these full models we selected the minimal model using a stepwise backward model selection procedure based on the lowest AIC values by removing factors with highest p-value in the model. If removing a variable resulted in a change of the AIC value of more than 2 and the resultant model was still parsimonious. e.g. following residuals diagnostics in DHARMa [52], the simplified model was kept. This procedure was repeated until removing variables no longer improved the model.

To ensure the models described above were appropriate, we also explored the use of alternative families such as poisson and quasipoisson, but DHARMa residuals diagnostics indicated that these failed to capture the dispersion in data. We tested for spatial autocorrelation both in

the data and residuals of the models using the Moran's I test [53] and found no significant spatial autocorrelation (for a p-value of 0.05), hence inclusion of a spatial term was not required. Finally, we estimated the correlation among our potential covariates, if two variables were >50% correlated, one of them was excluded from the final full model. These included temperature and relative humidity, immatures total number and larvae number, number of residents and number of children. Since larvae and immatures were correlated (53%), for the adult model we used the covariate 'immatures' (sum of larvae and pupae).

To estimate the overlap between the spatial distributions of adult and immature stages, we used the "nicheOverlap" function in the R package 'dismo' [54], which estimates an index of similarity between rasterized density distributions based on [55], and calculated the Pearson correlation coefficient between these two distributions using the 'layerStats' function in the package 'raster' [56]. The density maps were built based on the coordinates of each sampling location using the R package leaflet [57] which uses open-source basemaps from OpenStreetMap (CC BY-SA 2.0).

To analyse whether the location of blood-feeding (indoor vs outdoors) was in line with collection densities, the expected numbers of indoor blood fed females were predicted from the total bloodfed collections in each year and locality multiplied by the relative indoor density. A chi-square goodness of fit test was used to determine whether observed values deviated from predictions.

Other comparisons between proportions used chi-square contingency table tests or Fisher exact tests (depending on the expected values). *Stegomyia* index results were compared among localities using non-overlapping confidence intervals as indicative of a significant difference.

## Results

### Characteristics of sampled houses

A total of 1,163 houses were sampled in 2016 and 631 in 2017, plus an additional 24 public spaces in 2017 (S1 Table), including among others, places of worship, schools, market stalls and stores. All houses in 1200 LG (urban) were cement block buildings of modern designs, whereas the houses in Tabtenga (peri-urban) and Goundry (rural) were a mix of traditional adobe houses and modern houses. Average occupancy rates in each locality in 2016 were 5.3, 5.0 and 2.9 residents per house, and insecticide-treated nets (ITNs) were seen in 75.9%, 91.5% and 75.9% of houses, with an average of 0.44, 0.43 and 0.43 ITNs per person, respectively. In 2017, the average occupancy rates in each locality were 5.2, 6.2 and 3.3, and ITNs were seen in 60.6%, 87.8% and 88.7% respectively, with an average 0.36 ITNs per person in each locality.

### Adult mosquito species abundance

A total of 47,255 adult mosquitoes were collected during both years in all localities by indoor and outdoor resting catches (Table 1). *Aedes aegypti* abundance varied between the sampling localities in both years (2016: $\chi^2_2$ = 187.1; P<<0.001; 2017: $\chi^2_2$ = 165.2; P<<0.001). Adult *Ae. aegypti* were more abundant in 1200 LG and Tabtenga, where they comprised between 9.3 and 12.8% of the total catch, at house infestation rates of between 63.1 and 77.2%. Collections in Goundry were consistently lower than in the urban and peri-urban sites, comprising less than 5% of the total number of mosquitoes caught, with house infestation rates of less than 20.8%. Low numbers of other *Aedes* species were collected, mainly in Goundry, and included *Aedes vittatus*, *Aedes hirsutus*, and *Aedes metallicus* (Table 1).

Catches of other mosquitoes were dominated by *Culex quinquefasciatus* and *Anopheles gambiae s.l.* (Table 1). *Culex quinquefasciatus* was by far the most abundant mosquito in the two urbanised sites, 1200 LG and Tabtenga, where it comprised approximately 80% of the total

**Table 1. Species, number and proportion of adult mosquitoes collected indoors and outdoors in three localities of Ouagadougou in 2016 and 2017.** Proportions were calculated only for the most common vector species *Aedes aegypti*, *Anopheles gambiae*, *Culex quinquefasciatus*.

| | 1200LG (Urban) | | Tabtenga (Peri-urban) | | Goundry (Rural) | |
|---|---|---|---|---|---|---|
| **Species** | **2016** | **2017** | **2016** | **2017** | **2016** | **2017** |
| *Aedes aegypti* | 1811 (11.11%) | 782 (12.56%) | 976 (9.33%) | 963 (12.80%) | 137 (3.75%) | 143 (4.56%) |
| Other *Aedes* [1] | 17 | 1 | 8 | 3 | 35 | 46 |
| *Anopheles gambiae s.l.* | 585 (3.59%) | 143 (2.30%) | 601 (5.74%) | 237 (3.15%) | 2,107 (57.72%) | 1,659 (52.87%) |
| *Anopheles arabiensis* | 229 (96.22%) | 66 (92.96%) | 253 (92.67%) | 144 (92.90%) | 186 (34.90%) | 148 (15.24%) |
| *Anopheles coluzzii* | 7 (2.94%) | 4 (5.63%) | 13 (4.76%) | 10 (6.45%) | 320 (60.04%) | 746 (76.83%) |
| *Anopheles gambiae* | 2 (0.84%) | 1 (1.41%) | 7 (2.57%) | 1 (0.65%) | 27 (5.06%) | 77 (7.93%) |
| Other *Anopheles* [2] | 3 | 1 | 6 | 2 | 66 | 132 |
| *Culex quinquefasciatus* | 13,877 (85.13%) | 4,244 (68.14%) | 8,864 (84.71%) | 5,834 (77.55%) | 1,128 (30.90%) | 507 (16.16%) |
| Other *Culex* | 8 | 1047 | 6 | 477 | 140 | 574 |
| *Lutzia tigripes* | 0 | 8 | 0 | 7 | 36 | 30 |
| *Mansonia sp.* | 0 | 2 | 1 | 0 | 0 | 1 |
| Total collected | 16,301 | 6,228 | 10,462 | 7,523 | 3,649 | 3,092 |

[1] Other *Aedes* were *Ae vittatus*, *Ae hirsutus* and *Ae metallicus*

[2] *Anopheles* were *An. rufipes*, *An. funestus* and *An. ziemanii*. The proportions of *Anopheles gambiae* complex members were calculated using the total number of *An. gambiae s.l.* identified to species (rather than total collected) as denominator.

collections. In rural Goundry, numbers of *Cx. quinquefasciatus* were significantly lower, comprising less than 25% of catches ($\chi^2_1 = 191.9$, P<<0.001). In contrast, the proportions of *An. gambiae* s.l. in catches at Goundry were over ten times greater than at the urban and peri-urban localities (56% *vs.* 4%; $\chi^2_1 = 11.34$, P<<0.001). Molecular identification of *Anopheles gambiae* complex mosquitoes revealed significant variation in species composition ($\chi^2_4 = 1027$, P<<0.001) with the highest proportions *of An. coluzzii* seen in Goundry, whereas *An. arabiensis* was the dominant species in Tabtenga and 1200 LG (Table 1). Other *Anopheles* species (*Anopheles rufipes*, *Anopheles funestus* and *Anopheles ziemani*) and the predatory culicine *Lutzia tigripes* were recorded much more frequently in Goundry (1.2% vs 0.1%; Table 1).

### Resting location and diurnal activity of adult Aedes aegypti

Significantly more adult female *Ae. aegypti* were collected resting outdoors than indoors in all localities in each year (S2 Table), with a remarkably consistent proportion outdoors each year (mean = 0.73 in both 2016 and 2017), equivalent to an outdoor: indoor ratio of 2.7-fold (binomial test P<0.001). As can be seen in Fig 1 differences in relative proportions caught indoor and outdoor varied among localities ($\chi^2_2 = 149$, <0.001), with a consistently lower, but still strongly exophilic-biased, ratio in Tabtenga.

However, the proportion of bloodfed females caught outdoor was only slightly greater than those indoor (ratio = 1.05), which represents approximately twice the expected number predicted from the total indoor: outdoor catch ratio (Table 2). This suggests that a preference for exophily may not be sustained through the entire gonotrophic cycle.

Trends in morning *vs.* afternoon collections of *Ae. aegypti* were inconsistent across years (S3 Table). In 2016, there appeared to be a bias toward morning collections (59.6%; binomial test, P<0.001), but in 2017 morning and afternoon collections were very similar (50.3%; binomial test, P = 0.83). Similarly, strong variation between localities in morning:afternoon collections was evident in 2016, with relatively morning–biased collections in the urban and peri-urban sites but afternoon-biased in Goundry ($\chi^2_2 = 497$, P<0.001), but was barely-evident in 2017 ($\chi^2_2 = 6.1$, P = 0.047).

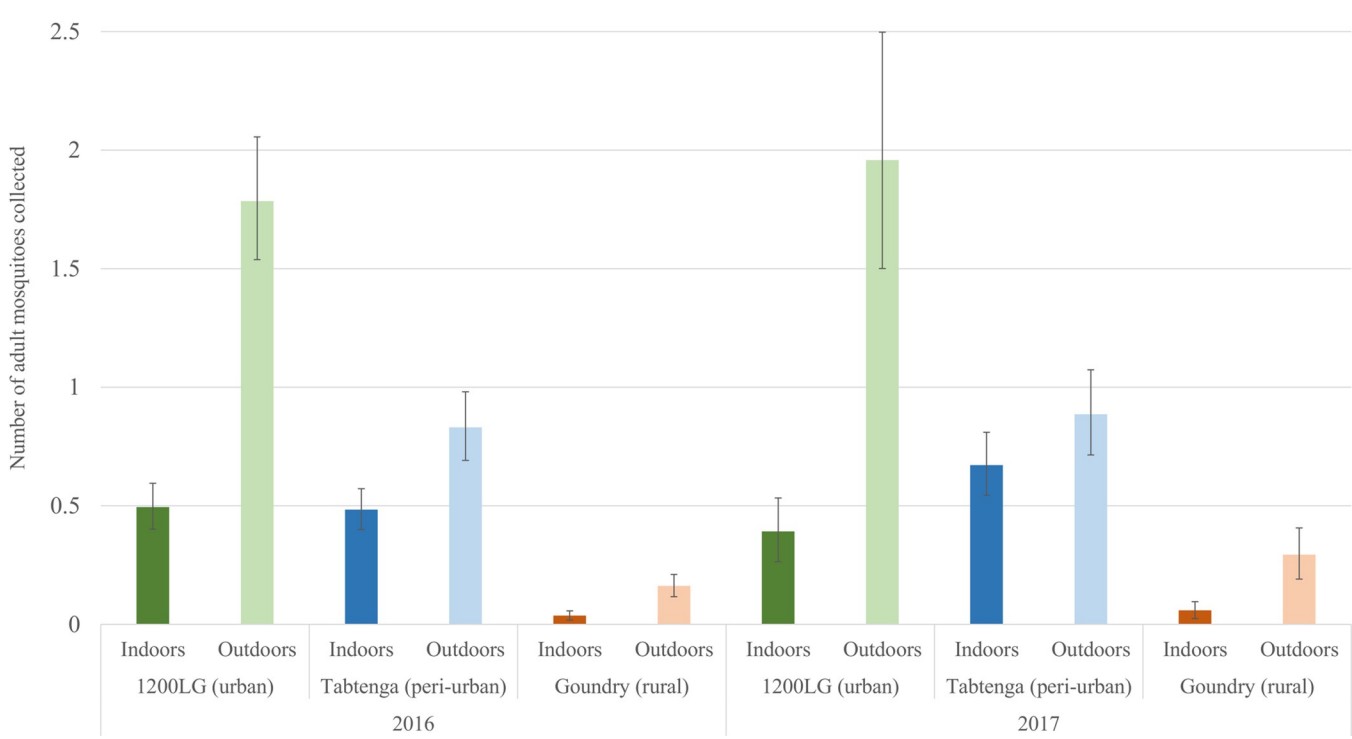

**Fig 1. Indoor and outdoor resting behavior of adult female *Ae. aegypti* in domestic housing in Ouagadougou.** Geometric means and 95% confidence limits of numbers of *Ae. aegypti* adults collected indoors and outdoors per house, in the three localities in 2016 and 2017.

## Host preference of adult Aedes aegypti

In total, 158/180 (88%) (S4 Table) of *Ae. aegypti* mosquito blood meal samples that amplified successfully by PCR were of human origin, with a higher proportion in 2016 (92%) than in 2017 (76%) ($\chi^2_1$ = 7.9, P = 0.005). None of the relatively few bloodfed mosquitoes from Goundry amplified successfully in 2017, but there was no significant difference in bloodfeeding rates between sites in 2016 ($\chi^2_2$ = 0.2, P = 0.90). Six bloodmeals contained blood from multiple hosts: five had human and canine blood and one contained both canine and bovine blood (S4 Table).

**Table 2. Numbers of bloodfed *Aedes aegypti* and density/house in indoor and outdoor resting collections in domestic housing in Ouagadougou.** The expected number of bloodfed mosquitoes indoors was calculated from the proportion caught indoors and compared to observed numbers bloodfed using a chi-square goodness of fit test.

| Year | Locality | Observed blood fed | | | | Observed density | | | | Proportion caught | Expected bloodfed | |
|---|---|---|---|---|---|---|---|---|---|---|---|---|
| | | Indoor | Outdoor | total | % indoor | Indoor | Outdoor | total | % indoor | Indoor | Indoor | X2 |
| 2016 | 1200LG | 118 | 186 | 304 | 39 | 0.51 | 1.79 | 2.30 | 22 | 0.22 | 67.41 | 37.97 |
| | Tabtenga | 110 | 92 | 202 | 54 | 0.48 | 0.83 | 1.31 | 37 | 0.37 | 74.02 | 17.50 |
| | Goundry | 5 | 13 | 18 | 28 | 0.04 | 0.16 | 0.20 | 20 | 0.20 | 3.60 | 0.54 |
| 2017 | 1200LG | 42 | 59 | 101 | 42 | 0.37 | 1.86 | 2.23 | 17 | 0.17 | 16.76 | 38.02 |
| | Tabtenga | 111 | 55 | 166 | 67 | 0.67 | 0.89 | 1.56 | 43 | 0.43 | 71.29 | 22.11 |
| | Rural | 3 | 4 | 7 | 43 | 0.05 | 0.28 | 0.33 | 15 | 0.15 | 1.06 | 3.55 |
| | | | | | | | | | | | total | 119.69 |
| | | | | | | | | | | | df | 5 |
| | | | | | | | | | | | P | <<0.001 |

### Predictors of Aedes aegypti *adult mosquito abundance*

Significant factors predicting *Ae. aegypti* adult abundance are shown in Table 3. Consistent with the analyses presented above, locality (urban, peri-urban, rural) was a strong determinant, as was the collection location (indoors *vs.* outdoors), with variation in the indoor:outdoor ratio between localities shown by the significant interaction term. Whilst collections in Tabtenga and Goundry were quite consistent between years, the number of *Ae. aegypti* collected in 1200 LG was much lower in 2017 than 2016 (evident in the significant year and year x locality interaction terms). However, it should be noted that the proportion of *Ae. aegypti* in the total mosquito catch was similar, and actually slightly higher, in 1200 LG in 2017 as a result of a much lower abundance of *Cx. quinquefasciatus* (Table 1). Collection month and rainfall also exerted significant effects with reduced abundance in October, and a positive relationship with rainfall over the past 14 days. Importantly, adult abundance was also predicted by the abundance of immature stages from containers in or around the same household, with larval and pupal collections pooled due to low pupal numbers. House type was retained in the minimal model, but explained little variation, whilst other factors did not add to predictive value and were excluded from the final model.

### Immature mosquito species abundance

*Aedes aegypti* comprised over 85% of immature stages collected from habitats in the urban and peri-urban localities, 1200 LG and Tabtenga, but only 46.4% in rural Goundry. In contrast,

**Table 3. *Aedes aegypti* adult model glmm showing predictors beta estimates of effect size.** Confidence intervals, test statistic (z-value) and associated probability for the minimal model. Significant predictors are highlighted in **bold text**, and non-significant terms, not included in the model are listed on the bottom line.

| Predictors | Beta Estimate | 95%CL | z-value | Pr(>|z|) |
|---|---|---|---|---|
| **(Intercept)** | **-4.64** | **[-5.62–3.66]** | **-9.28** | **<0.001** |
| **Year [2016]** | **0.99** | **[0.5–1.48]** | **3.95** | **<0.001** |
| Locality [Rural] | | | | |
| **Peri-urban** | **3.56** | **[2.63–4.5]** | **7.45** | **<0.001** |
| **Urban** | **4.58** | **[3.55–5.61]** | **8.70** | **<0.001** |
| Location[indoors] | | | | |
| **outdoors** | **1.77** | **[1.31–2.22]** | **7.67** | **<0.001** |
| Month [August] | | | | |
| **October** | **-0.91** | **[-1.45–0.38]** | **-3.34** | **<0.001** |
| September | 0.04 | [-0.28–0.35] | 0.23 | 0.817 |
| **14 days rainfall** | **0.00** | **[0.00–0.01]** | **2.28** | **0.023** |
| House type [Mixed] | | | | |
| Modern | -0.36 | [-0.73–0.01] | -1.90 | 0.058 |
| Traditional | -0.42 | [-1.23–0.39] | -1.01 | 0.311 |
| **Immature abundance** | **0.00** | **[0.00–0.01]** | **4.90** | **<0.001** |
| Year [2016]: Locality [Rural] | | | | |
| Year: Peri-urban | -0.56 | [-1.13–0.00] | -1.94 | 0.052 |
| **Year: Urban** | **-1.19** | **[-1.78–0.59]** | **-3.91** | **<0.001** |
| Locality [Rural]: Location [indoors] | | | | |
| **Peri-urban: outdoors** | **-1.34** | **[-1.84–0.85]** | **-5.31** | **<0.001** |
| Urban: outdoors | -0.28 | [-0.8–0.24] | -1.04 | 0.297 |

Non-significant terms were collection time(am/pm), house type, children number, ITNs presence ITNs number, animals presence, animal number, locality*year, Locality*Location

Confidence intervals, test statistic (z-value) and associated probability for the minimal model. Significant predictors are highlighted in **bold text**, and non-significant terms, not included in the model are listed on the bottom line.

*Aedes vittatus*, which comprised less than 0.5% of the immatures collected in the two urban localities, amounted to 40.6% of the total in Goundry (S5 Table). *Aedes vittatus* immatures were found mainly in water residues in the animal drinking troughs that were more common in Goundry than in the urban sites (S1 Fig). Few immature *Anopheles gambiae* s.l. were collected in any site, as would be expected for a species that typically breeds in ground water pools rather than the containers sampled in this study.

Overall mosquito community composition was strongly influenced by collection location, which was consistent across the two sampling years, with significant difference between rural Goundry and the two urban sites (S1 Fig).

## Habitats of immature stage Aedes aegypti

Across all three localities, a total of 1,445 containers were inspected during the study of which 666 (46%) contained *Aedes aegypti* larvae or pupae (Fig 2). Immatures were found in all container types inspected and although infestation rates of container types were significantly different across the localities ($\chi^2$ = 170; P<<0.001; Fig 2), there were some consistencies. Tires were among the most heavily infested in both urbanised localities, reaching rates of 31% and 32% in Tabtenga and 1200 LG respectively, but they were of minimal importance in Goundry, where infested domestic water storage drums (40%) were the most heavily infested (Fig 2).

Not all containers found to contain larvae may support development to the pupal stage and be regarded as productive for breeding. However for pupae, in urban and peri-urban sites, tires were highly productive habitats, containing 37% and 34% of all pupae in 1200 LG and Tabtenga respectively (Table 4 and Fig 2).

Other highly productive containers included large water storage containers (drums, jars or barrels) from which, in Tabtenga, an area without piped water, 40% of pupae were found. In contrast, these large containers produced only 13% of pupae in 1200 LG, an area with piped water, and where instead, small containers including miscellaneous objects (plastic and metallic boxes, terracotta pots, plastic shoes,..) were responsible for nearly 40% of the vector population.

Large water drums or barrels were also important in the semi-rural locality Goundry (30%), where the most productive habitats were animal water troughs (44%).

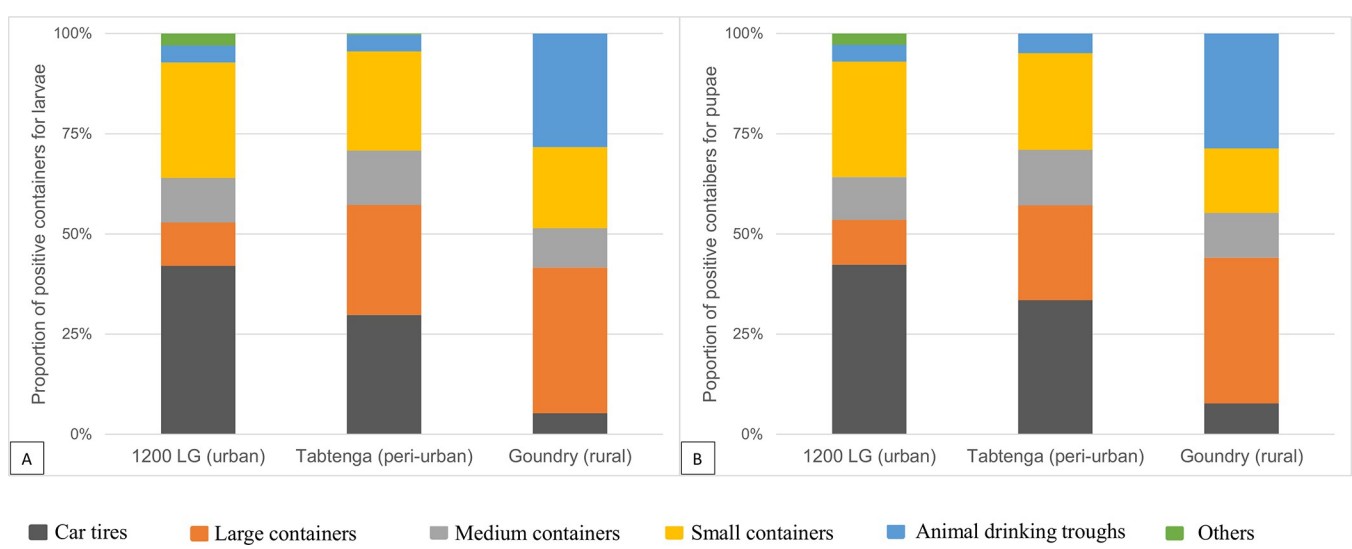

**Fig 2. Distribution of all larvae (Fig 2A) and pupae (Fig 2B) across the different positive container types.**

**Table 4. Numbers of *Aedes aegypti* pupae found infesting containers and other potential habitats by locality, year of sampling and habitat type.**

| | 2016 | | | 2017 | | | | |
|---|---|---|---|---|---|---|---|---|
| | **Aug** | **Sep** | **Oct** | **Aug** | **Sep** | **Oct** | **No. pupae** | **% of total pupae** |
| 1200 Logements | | | | | | | | |
| Tires | 277 | 322 | 17 | 219 | 118 | | 953 | 36.91 |
| Large containers | 31 | 117 | 68 | 99 | 21 | | 336 | 13.01 |
| Medium containers | 63 | 37 | 0 | 40 | 10 | | 150 | 5.81 |
| Small containers | 255 | 470 | 14 | 249 | 31 | | 1019 | 39.47 |
| Animal troughs | 16 | 19 | 0 | 6 | 32 | | 73 | 2.80 |
| *Others | 48 | 0 | 0 | 3 | 0 | | 51 | 2.00 |
| total | 690 | 965 | 99 | 616 | 212 | 0 | 2582 | 100.00 |
| Tabtenga | | | | | | | | |
| Tires | 404 | 373 | 3 | 281 | 365 | | 1426 | 33.98 |
| Large containers | 154 | 362 | 0 | 365 | 250 | | 1131 | 26.95 |
| Medium containers | 219 | 432 | 9 | 96 | 35 | | 791 | 18.85 |
| Small containers | 91 | 377 | 39 | 139 | 45 | | 691 | 16.47 |
| Animal troughs | 29 | 81 | 1 | 46 | 0 | | 157 | 3.75 |
| *Others | 0 | 0 | 0 | 0 | 0 | | 0 | 0.00 |
| total | 897 | 1625 | 52 | 927 | 695 | 0 | 4196 | 100 |
| Goundry | | | | | | | | |
| Tires | 22 | 56 | 0 | 85 | 0 | 0 | 163 | 4.9 |
| Large containers | 644 | 179 | 0 | 102 | 48 | 31 | 1004 | 30.24 |
| Medium containers | 46 | 37 | 34 | 1 | 8 | 99 | 225 | 6.78 |
| Small containers | 139 | 72 | 0 | 261 | 1 | 0 | 473 | 14.25 |
| Animal troughs | 671 | 124 | 1 | 541 | 88 | 30 | 1455 | 43.83 |
| *Others | 0 | 0 | 0 | 0 | 0 | 0 | 0 | 0.00 |
| total | 1522 | 468 | 35 | 990 | 145 | 160 | 3320 | 100 |

*Others includes ground water puddles, tree holes and plant pots of less than 3*L*.

## Predictors of *Aedes aegypti* larval and pupal densities

A generalised linear model fitted to the number of *Aedes aegypti* larvae collected per breeding site showed an effect of year (2017 collections > 2016 collections) and significant differences between localities, which were consistent across years, with both the urban and peri-urban localities having much higher densities than the rural site, Goundry (S6 Table). Container type influenced larval density with highest larval densities found in tires. Higher water levels in containers were also associated with higher larval density and highest larval densities occurred in September compared to August and October. Factors such as cumulative rainfall totals measured over 2 or 7 days, temperature, container purpose, number of residents, and numbers of adult mosquitoes collected had no significant associations with larval density.

Pupal density did not differ significantly between years, and was higher only in Tabtenga than Goundry, with urban 1200 LG not significantly different (S7 Table). Container type influenced pupal density, but in contrast to larvae, tires were not significantly more productive, with only animal drinking troughs significantly higher than the reference category. Container utility was important as pupal density was reduced by 43% in functional containers compared with non-functional/discarded containers Pupal density was also negatively associated with mean temperature and was also positively associated with the number of adult mosquitoes collected in the same house.

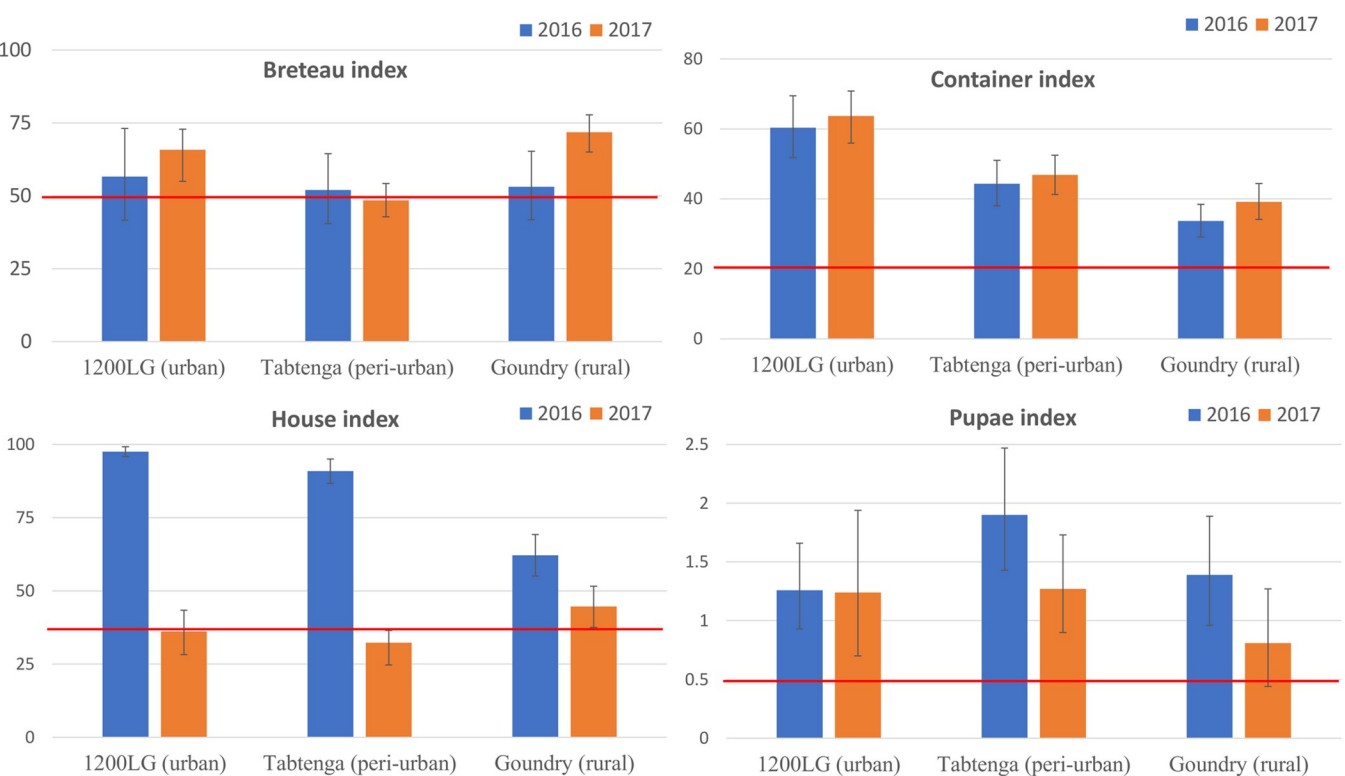

**Fig 3. Breteau, container and houses indices (and 95% confidence intervals) and the number of pupae per person (PPI) (and 95% confidence limits) per year and per locality.** The red line indicates the WHO thresholds that are set at 50, 20 and 35 respectively for BI, CI and HI. The PPI threshold is estimated based on an initial seroprevalence of 33% and an average temperature of 28°C and for an increase of 10% of the seroprevalence [37].

### Stegomyia *and pupae/person indices*

The *Stegomyia* indices are summarized in Fig 3, and show that the WHO thresholds were exceeded for all stegomyia indices in Goundry. In the peri-urban locality of Tabtenga, the container and pupal indices exceeded the WHO thresholds, but the Breteau and House Indices, were less informative in 2017.

### Spatial distribution of immature stages and adult collections

Mapping of immature and adult densities showed some, albeit very imperfect overlaps between the different life stages in each locality (Fig 4). Niche overlap analysis showed that approaching half of the distribution of adults and immatures overlapped (index = 0.44). Whilst this varied between locations, with a lower index value in Goundry (index = 0.29) compared to peri-urban (index = 0.56) and urban (index = 0.55) the overall spatial correlation of the distribution is consistently high (Pearson correlation = 0.74) regardless of the location.

## Discussion

The objective of this study was to generate detailed baseline data on the biology and behaviour of *Ae. aegypti* in Burkina Faso, contributing to the essential evidence-base for developing dengue prevention and outbreak plans. The key findings indicate that the arbovirus vector *Ae. aegypti* is common throughout Ouagadougou, and most abundant in the highly populated central areas where infestation rates reach 78% of dwellings. Adult females are predominantly

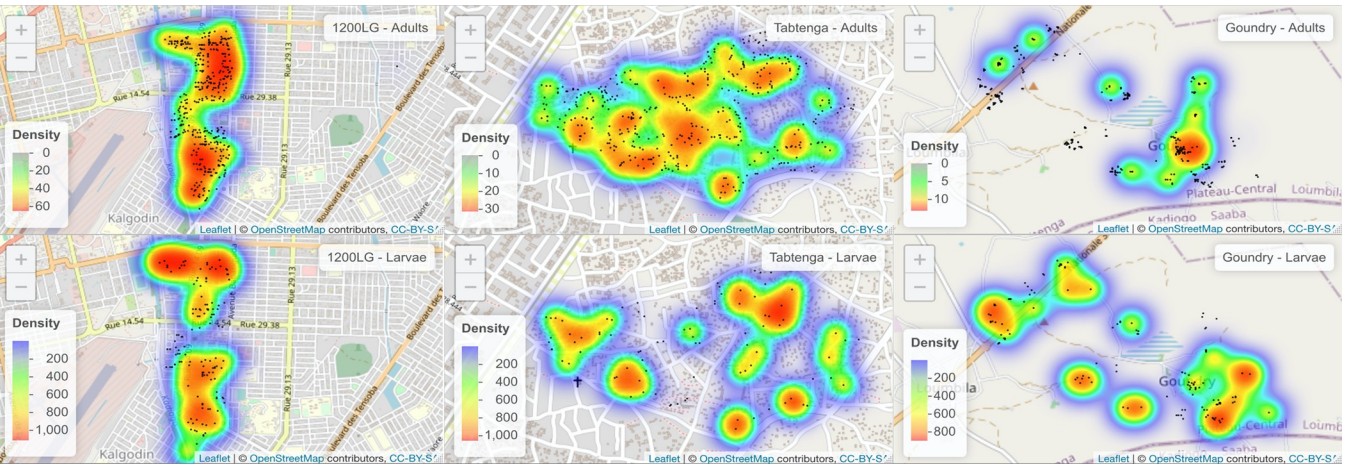

**Fig 4. Density maps of *Aedes aegypti* adult (top) and larval (down) densities per house in Goundry (left), Tabtenga (middle) and 1200LG (right).** The dark black spots represent the houses that were sampled. The base map was downloaded from https://rstudio.github.io/leaflet/., which follows the terms of use in https://www.openstreetmap.org.

anthropophagic and also highly exophilic, though feeding more indoors than exophily rates would suggest. Females also appear to oviposit in all container types, both in use or discarded.

## Typology of breeding sites

The key container habitats, those harboring the highest numbers of pupae, and from which the greatest numbers of adults emerge, were discarded car tires, large domestic water containers (drums and barrels) and small containers (including discarded vessels). *Aedes aegypti* were caught in both mornings and evenings, consistent with their expected pattern of diurnal activity, but it is unclear to what extent they may also be nocturnally active.

The profile in the rural outskirts of the city appears to be quite different. Here there was a greater diversity of mosquito species, the most common of which were *Anopheles gambiae* and *Culex quinquefascia*tus, and *Aedes vittatus*. The most productive containers in Goundry for both *Ae. aegypti* and *Ae. vittatus* were animal drinking water troughs. The most significant mosquito-borne disease risk here is malaria, though the presence of significant numbers of *Ae. vittatus* is of note pending more insight into any vectorial role it might have in this area. *Aedes vittatus* occurs throughout Africa, tropical Asia, and southern Europe and was recently discovered in the Caribbean [58]. Although its vectorial capacity is not known, it occurs in both sylvatic and peridomestic environments, feeds on humans and is involved in the maintenance and transmission of yellow fever, Zika, chikungunya, and dengue [59].

However, in the higher density urban and peri-urban areas within Ouagadougou, *Ae. aegypti* was the only arbovirus vector of concern and was, almost certainly, the only active vector during the dengue outbreaks of 2016 and 2017. The likelihood of future outbreaks of dengue and other arboviruses transmitted by *Ae. aegypti* is high and a programme to prevent, limit or respond to dengue outbreaks must now be considered a public health priority to protect urban populations.

Ideally, dengue vector control programmes should involve two strategies, the first for vector population suppression, a year-round or pre wet season programme to maintain a low vector population, and the second strategy for responding rapidly to disease outbreaks[60]. Suppressing vector populations typically employs sustainable appropriate methods targeting the immature stage habitats. Diligent sustained action by community groups can have major impacts,

and be effective in both outbreak prevention and response, even in the absence of insecticides [61–64]. This requires regular waste disposal for elimination of discarded materials, containers with no function and general garbage. Water barrels, tanks or jars or any other functional or in-use containers should have tightly fitting lids, while empty bottles and buckets should be stored inverted. Targeting the highly productive container types such as those that produce over 70% of all *Ae. aegypti* pupae, is recommended by WHO [28]. Although more work is required to identify all key container types in Ouagadougou in dry and wet seasons, it is already clear that car tires are an important habitat responsible for producing over 30% of adult vectors in the wet season, with drums and barrels used for water storage potentially increasing in importance in the dry season. Tires are a unique and easily recognised waste item, apparently with no monetary value and a successful campaign to remove them would be a useful milestone (while raising the difficult question of what to do with the tires collected). Before decisions are reached however, the survey should be repeated in the dry season, when the vector population will be at its lowest and most vulnerable level and the most productive container types might be different.

Larviciding can also be used for larval reduction using temephos, an organophosphate and two biological insecticides Bti and pyriproxyfen. Preliminary data shows organophosphates are effective against *Aedes aegypti* larvae in Burkina [65], but data on efficacy of Bti and pyriproxyfen are awaited. Although the identification of key containers for pupal productivity may reduce the challenge, the diversity and the number of breeding containers will compromise larviciding as a stand-alone method for dengue control in Burkina Faso.

## Aedes aegypti *resting and blood feeding behavior*

*Aedes aegypti* adults can be prevented from entering buildings by screens fitted to windows, which do not necessarily need to be insecticide-treated to be effective [66,67]. Indoor resting can be controlled by targeted indoor residual spraying (TIRS), where only the lower half of the walls are treated with insecticide or by using hand-held aerosol cans to spray known resting indoor sites [68–70]. Clearly, further work is required to fully elucidate resting preferences, a critical question for planning control, and the efficacy of IRS against *Ae. aegypti* in Africa need to be evaluated experimentally as a priority even if the exophilic behaviour may recommend additional control methods. The high levels of exophily recorded in this study (Fig 1) which doesn't exclude high exophagy levels in the same *Aedes aegypti* populations, raise the question of whether IRS is an appropriate method for control of *Ae. aegypti* in west Africa. Of the two *Ae. aegypti* forms known to occur in Burkina Faso, only *Ae. aegypti aegypti* (*Aaa*) exhibits synanthropic behaviour, including endophily and endophagy [71]. Endophilic behaviour has been documented in *Ae. aegypti* since the earliest research [72] and is a common feature of populations worldwide [73,74]. Throughout its range in Africa, substantial or even preferential outdoor activity by *Aaa* is not unusual, and has been reported from Ghana [75], Senegal [40] and Kenya [76]. Exophilic or endophilic preferences may not be exclusively one or the other, and many *Ae. aegypti* populations exhibit both. For example, in Kenya, Teesdale recorded details of *Ae. aegypti* daily cycles of movement in and out of houses [76], while a study in Mexico recorded indoor and outdoor biting and resting, with exophagy the more common [77]. Despite this, the studies demonstrating the impact of TIRS (Targeted Indoors Residual Spray) on *Ae. aegypti* were also performed in Mexico. In Burkina Faso, a significantly higher than expected proportion of bloodfed females are found resting indoors, which suggests that exophily may not dominate the entire adult stage and that adult females are likely endophagic or come indoors at some stage after bloodfeeding.

The majority of bloodmeals were identified as human in origin, with the remainder from dogs and only one sample from cattle (S4 Table). Notably, only 6–7% of bloodmeals were non-

human in samples from the urban localities, but 28% of bloodmeals in the semi-rural site were from dogs. *Aedes aegypti* exists in two forms in Africa, *Ae. aegypti aegypti* (*Aaa*) inhabit domestic environments, breed in artificial containers, and are highly anthropophagic, and *Ae. aegypti formosus*, (*Aaf*) the forest form which is zoophilic [78]. The abdominal scaling patterns used to discriminate the forms [42,79] were not found useful for discriminating forms among our samples from Ouagadougou [71]. Genetic studies have shown that the Ouagadougou population is an interbred population mixture of *Aaa* and *Aaf*, expressing intermediate animal and human preferences [4]. The results presented here point towards an urban population displaying predominantly *Aaa* behaviour and a rural population that is still anthropophilic but with a far greater likelihood of zoophagy [4]. Clarification of host preference is an important element of a mosquito population's vectorial capacity but doing so will require a larger sample size and additional studies to gain more insight. This should be prioritised together with the studies on indoor/outdoor feeding and resting preferences.

## Mosquito species diversity

Higher densities of adult *Ae. aegypti* were found in urban and peri-urban localities of 1200 LG and Tabtenga, compared to Goundry, the rural locality. Adults rested mainly outdoors in all sites, feeding preferentially on humans with rare canine or bovine bloodmeals and their densities were affected by month and the year of collection, the locality, the indoors/outdoors location and at a lesser extent by the immature stage abundance and the cumulative rain of 14 previous days.

Ae. *aegypti* was the main *Aedes* species collected at all developmental stages, in all localities, in both years. *Ae. aegypti* adults and larvae densities followed a negative gradient from urban to rural localities. Urbanisation has been identified as the main driver of *Ae. aegypti* proliferation in Africa [4,80], and other environmental changes resulting from human activities promote higher abundance and lower species diversity; lower abundance and higher species diversity are more typical of natural environments and ecosystems [81]. In our study, the diversity of all culicines was greater in the rural site, Goundry, than in the urban and peri-urban localities. Goundry is predominantly agricultural land at the edge of the bush, with trees and scrubland beyond. *Aedes vittatus* was common here only, preferring animal drinking troughs as larval habitats. Also found only at Goundry, the predatory larvae of *Lutzia tigripes* shared some habitats with a prey species, *Ae. aegypti* and may have contributed to the lower densities of *Ae. aegypti* in Goundry compared to Tabtenga and 1200 LG. Important vector mosquitoes included *Culex quinquefasciatus* which was common here though not as abundant as at the other localities, and *Anopheles gambiae* s.l., which was common.

## Determinants of larval and adult mosquito densities

Multiple containers types were found to contain larvae in urban and peri-urban sites, with tires the most common. The typology of containers may vary according to the locality. Drums and barrels that are used for water storage, are more abundant in the rural and peri-urban localities of Goundry and Tabtenga where piped water is either absent or rare. Studies in central and East Africa have reported tires as the main breeding sites for *Aedes* mosquitoes [80,82,83]. In Indonesia, the most abundant breeding containers were bird watering dishes, tires had the highest frequency of positivity, while large open tanks storing water (known as *Bak Mandi*) were the most productive containers [84]. The relative importance of any container is highly context-specific and any container's contribution to the vector population can easily be underestimated [85,86].

Breeding site characteristics that affect immature stages abundance and adult life history traits include among others, dissolved oxygen, water temperature, pH, conductivity and

salinity [87]. Characteristics such as dissolved solids, ammonia, nitrate, and organic matter vary significantly between urban and rural containers, which might explain some urban-rural differences in breeding of *Ae. aegypti* [88]. We examined a limited number of breeding sites characteristics and environmental variables and found that container types and water levels within, can increase larval density while containers that are in use, or classed as useful, decrease pupal density. Cumulative rainfall of the previous 14 days and mean temperature affect adult and pupal densities respectively. Investigations in Iquitos used a generalised additive model to highlight the important contribution to *Ae. aegypti* adult density of weather-related covariates including temperature, rainfall and wind [89]. Though our study did not consider other covariates related to breeding sites, the density of immature stages was the strongest covariate in the model contributing to adult density.

Immature stages as well as adult densities were more affected by the month of collection with September the peak month of higher densities, compared to August and October. Although immature stages contributed to adult density, only pupal density was affected by adult abundance. The spatial distribution of adult and immature stage hotspots showed consistent overlapping and limited dispersal of adults from the immature stage location as shown in Lacon et al [90] and in Bonnet et al. [91] who showed that removing breeding sites had a direct impact on proliferation of adults. The lower overlap in adult *vs.* larval densities in the rural compared to urban locations might be explained by a closer proximity of vegetation around households in the former, which offer potentially suitable, but difficult to sample, resting places for adults.

## Need for context-specific new tools for dengue risk estimation

As the Stegomyia indices have not previously been reported from Burkina Faso, we do not know what levels are typical during a non-outbreak period. During our study, the Stegomyia index values exceeded the WHO threshold for dengue risk in the rural locality of Goundry; paradoxically as this was the site where the lowest numbers of adult *Aedes aegypti* were recorded, and far from the central area of Ouagadougou where mainly urban and cases were known to be concentrated. In addition, there is no evidence that any quantifiable associations exist between these indices of vector immature stages abundance and dengue transmission, and the thresholds have little value for prediction of dengue outbreaks [92,93]. Instead, research has moved to investigating whether numbers of adult female *Ae. aegypti* numbers can provide more accurate and reliable alerts [93].

In this study the number of total adults collected was highly correlated with the number of fed females collected and their density models shared the same explanatory variables. In contrast to *Stegomyia* indices, which are based on immature stage numbers, estimating the total number of adult *Ae. aegypti* can be more informative but remains challenging, with precision depending on the sampling methods and sampling efforts [37]. A pattern of similarities between spatial distribution of adults and immature stages are found, with hotspots overlapping in the urban sites, at least. More consistent indices are needed, to take into account *Ae. aegypti* resting and blood feeding behaviour, and the typology of breeding sites that are productive for pupae, yet potentially specific to the locality. Associating dengue active case detection in the community with holistic collection of *Ae. aegypti* bionomics collection could allow more consistent inferences to be made.

## *Perspective for* **Aedes aegypti**-*transmitting disease control*

Without vaccines for three of the four arboviruses transmitted by *Ae. aegypti*, Burkina Faso, like every other country, must rely on existing vector control tools as it plans its dengue control

programme. Burkina Faso has endured a heavy malaria burden for decades during which time it built considerable capacity in vector control of *Anopheles sp*. [94]. It is now attempting the same for *Ae. aegypti*, a mosquito with very different biology and behavior to malaria vectors and consequently, requiring a different approach for control.

Since so many larval habitats are essentially waste materials, clean-up campaigns accompanied by appropriate education and information programs are essential and, in fact have already been established in Ouagadougou [91]. However, selection and likelihood of success of even the most appropriate interventions should be based on more than method of delivery and insecticide susceptibility of the target population. Gaining access to a home to deliver indoor treatments may be prevented if occupants are out working or studying elsewhere. If this type of 'refusal' occurs at a high rate, achieving satisfactory coverage would be compromised. A house's structure, shape or construction materials can determine its suitability for fitted interventions such as window screens. Despite evidence that it is rarely effective [95], fogging or space-spraying outdoors is a popular response to *Ae. aegypti* borne arbovirus outbreaks worldwide. A high-profile activity, space-spraying is routine for many local authorities worldwide, the expected response to an outbreak although it can impact on transmission only if it is applied at frequent intervals and sustained for many weeks [89,90]. The exophilic character of the *Aedes aegypti* populations in Ouagadougou may support this method but as the same vector populations also show high resistance to pyrethroid insecticides [65,96], selection of insecticide should be based on up to date insecticide susceptibility testing together with safety considerations.

Pending the clarification of the vector population's behavioral preferences in the key areas described above, a program for vector control of dengue in Burkina Faso should be possible despite the limited range of interventions available. The Wolbachia method offers unprecedented impacts [97] but a date when it might be considered affordable and biologically suitable for use in Africa, may be many years away. Until then, or until a time when another equally effective intervention method is available, existing methods should be sufficient to at least reduce the frequency and mitigate the impact of outbreaks.

## Conclusion

This report describes the *Ae. aegypti* population before and during dengue outbreaks in Ouagadougou. Although additional dry season data are required, the study provides the most complete contemporary description of an *Ae. aegypti* population in West Africa and provides sufficient evidence to develop programs for prevention and control of outbreaks. Recognizing that many of the breeding, bloodfeeding and resting site preferences reported here are likely to occur in vector populations elsewhere in West Africa, the importance of undertaking further research to characterize those habits in additional vector populations cannot be overly stressed. Together with insecticide resistance status, these preferences determine the success of any control method and where possible, characterization should be based on evidence rather than assumption, especially if based on contexts outside Africa.

This study has done little to alter the view that the Stegomyia indices have limited epidemiological value or that they are likely to be more relevant in Africa than they have been elsewhere. Identifying alternatives however remains elusive. While there are similarities between immature stage and adult densities and similarities in spatial distribution, determining how these might be applied or how they could be combined with additional epidemiological parameters to generate more accurate indices reflecting the transmission potential of *Ae. aegypti* and disease risk remains a challenge.

All of these topics fit well within a regional approach to arbovirus control. Networks mapping *Ae. aegypti* key behaviors across the African continent, together with accurate indices of

arbovirus risk, insecticide susceptibility/resistance status and key epidemiological parameters would be an important step towards an effective regional/ global control strategy. Such an initiative is suspected to have broad support.

## Supporting information

**S1 Table. Household characteristics and demography in the three localities during 2016 and 2017.**
(DOCX)

**S2 Table. Average number per house, 95% confidence limits [in brackets] and the total number of *Aedes aegypti* (in parenthesis) collected in the study localities in 2016 and 2017.**
(DOCX)

**S3 Table. Average number of mosquitoes per house, 95% confidence limits [in brackets] and the total numbers (in parenthesis) collected in the morning (am) and in the afternoon (pm) of the three main species of mosquito collected in the study localities in 2016 and 2017.**
(DOCX)

**S4 Table. Number of bloodfed, number of PCR-tested, and bloodmeal sources of *Aedes aegypti* mosquitoes per locality and per year.**
(DOCX)

**S5 Table. Relative abundance of mosquito species collected as larvae and pupae during routine house sampling in each locality and year.**
(DOCX)

**S6 Table. Generalised linear mixed model of *Aedes aegypti* larval density.**
(DOCX)

**S7 Table. Generalised linear mixed model of *Aedes aegypti* pupal density.**
(DOCX)

**S1 Fig. Mosquito community diversity in immature collections in each location and year.**
(DOCX)

## Acknowledgments

The authors thank members and leaders of communities in the localities of 1200 Logements, Tabtenga and Goundry for their permission to perform the study and their cooperation throughout.

## Author Contributions

**Conceptualization:** Athanase Badolo, David Weetman, Philip J. McCall.

**Data curation:** Athanase Badolo, Aboubacar Sombié.

**Formal analysis:** Athanase Badolo, Mafalda Viana, David Weetman, Philip J. McCall.

**Funding acquisition:** Athanase Badolo, David Weetman, Philip J. McCall.

**Investigation:** Athanase Badolo, Aboubacar Sombié, Félix Yaméogo, Dimitri W. Wangrawa, Aboubakar Sanon, Patricia M. Pignatelli, David Weetman, Philip J. McCall.

**Methodology:** Athanase Badolo, David Weetman, Philip J. McCall.

**Project administration:** Athanase Badolo, Antoine Sanon.

**Resources:** Athanase Badolo, Antoine Sanon, Hirotaka Kanuka.

**Software:** Athanase Badolo, Mafalda Viana.

**Supervision:** David Weetman, Philip J. McCall.

**Validation:** Athanase Badolo, Mafalda Viana, David Weetman, Philip J. McCall.

**Visualization:** Athanase Badolo, Mafalda Viana, David Weetman, Philip J. McCall.

**Writing – original draft:** Athanase Badolo.

**Writing – review & editing:** Mafalda Viana, David Weetman, Philip J. McCall.

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
