## [Decision Letter · Decision Letter 0]

20 Apr 2022

Dear Prof McCall,

Thank you very much for submitting your manuscript "Bionomics of Aedes aegypti during the 2016-2017 dengue outbreaks in Ouagadougou, Burkina Faso" for consideration at PLOS Neglected Tropical Diseases. As with all papers reviewed by the journal, your manuscript was reviewed by members of the editorial board and by several independent reviewers. The reviewers appreciated the attention to an important topic. Based on the reviews, we are likely to accept this manuscript for publication, providing that you modify the manuscript according to the review recommendations. 

Sincerely,

Joseph T. Wu

Deputy Editor

Joseph Wu

Deputy Editor

Reviewer's Responses to Questions

**Key Review Criteria Required for Acceptance?**

**Methods**

-Are the objectives of the study clearly articulated with a clear testable hypothesis stated?

-Is the study design appropriate to address the stated objectives?

-Is the population clearly described and appropriate for the hypothesis being tested?

-Is the sample size sufficient to ensure adequate power to address the hypothesis being tested?

-Were correct statistical analysis used to support conclusions?

-Are there concerns about ethical or regulatory requirements being met?

Reviewer #1: The objectives of the study are well articulated and hypothesis has been adequately tested. Appropriate statistical analysis has been performed to support the conclusions of this study.

Reviewer #2: The study analyses a large sample size and uses a number of different criteria. The methods are well suited to the aim of analysing population dynamics of a disease vector.

**Results**

-Does the analysis presented match the analysis plan?

-Are the results clearly and completely presented?

-Are the figures (Tables, Images) of sufficient quality for clarity?

Reviewer #1: The results have been clearly presented with good quality tables and images

Reviewer #2: The description of the results, tables and figures are comprehensive and of goog clarity. I have included minor corrections into my review report (see below/word file)

**Conclusions**

-Are the conclusions supported by the data presented?

-Are the limitations of analysis clearly described?

-Do the authors discuss how these data can be helpful to advance our understanding of the topic under study?

-Is public health relevance addressed?

Reviewer #1: The conclusions are supported ny the data presented and has been carefully outlined to guide the reader in understanding Ae. aegypti populations before and during dengue outbreaks in Ouagadougou

Reviewer #2: The authors are able to draw clear conclusion on the characteristics of the population, the impact of a number of external factors and the suitability of Stegomya indices. Concluding from their results, they discuss a number of public health measures that can be taken to control the Aedes population and prevent future dengue outbreaks.

**Editorial and Data Presentation Modifications?**

Reviewer #1: Editorial suggestions have ben included in the annotated manuscript.

Reviewer #2: Title: The title seems to be underselling the manuscript a little bit. One could emphasize that this is the first comprehensive analysis of a West African population

Abstract:

L32: punctuation missing after PCR

L36: Isn’t Culex the most abundant mosquito sampled? See also my comments on table 1.

Methods:

L119 ff: How do the housing densities in the two urban areas compare?

Results:

L296 ff: Table1 indicates that Culex is the most abundant in the adult collections, which is opposing the statement in the Abstract that Aedes agyptii is the most abundant one, or does that only refer to immature stages, as indicated in figure S1? If Culex is most dominant in the adult samples but Aedes in the immature ones, how are these differences explained?

Also on table 1: I assume the fractions given for the Anopheles gambiae species indicate their percentage in the species complex and not in the overall sample. This should be indicated. But why do the numbers of Anopheles gambiae species do not add up?

L325 ff: Table 2: remove ‘§’ from headline. On right hand side some numbers seem to have shifted? Under % indoor the fraction of 1 rather than the percentage is given, which could be misunderstood at first glance.

L434 ff: Is there any idea/explanation why the overlap of immature and adult distribution is notably lower in the rural area? Why are adults completely missing from some sites where larvae were sampled? 

L656 ff (Figure legends) The legends to figure 2 refers to figure 3 and vice versa, this needs to be corrected and the entire text should be checked if references to figures follow the right numbering

Also on figure legend 3 (referring to figure 2): The colour code is not given. In addition, the description ‘Proportion of positive containers for larvae’ does not fully fit what is shown. It seems to rather be a distribution of all larvae across the different container types, adding up to 100% in total

L667/ figure 4: I assume the dark black spots represent the houses that were sampled? This should be indicated in the figure legends

Discussion:

General: I would suggest the use of sub-headings in the discussion to improve structure and readability

**Summary and General Comments**

Reviewer #1: (No Response)

Reviewer #2: The manuscript by Badolo et al. characterizes the bionomics of the vector mosquito Aedes aegyptii in Burkina Faso in 2016 and 2017, and sampling coincided with dengue outbreaks in these years. Immature stages as well as adult mosquitos were collected and identified. In additions, the origin of the female bloodmeal of adult Aedes aegyptii females was determined using PCR. Findings include the characterization of a highly anthrophilic population with higher adult densities in urban than in rural location. Stegomyia indices proved less suitable indicators of adult populations, whilst other influences like container types and rainfall levels were stronger influences. The authors discuss their findings with respects to containment measures. 

The present study is based on a large sample size and the analyses have been done with great care. Hence it provides a very useful resource to understand the bionomics of West African Aedes populations in relation to disease outbreak. It does not include data on the population during non-outbreak years or during the dry season, which could be included in future studies. 

I recommend publication of the manuscript and I only have some minor comments on the text and figures, which are given below.

PLOS authors have the option to publish the peer review history of their article (what does this mean?). If published, this will include your full peer review and any attached files.

Reviewer #1: Yes: Dr. Clarence M. Mang'era

Reviewer #2: No

Figure Files:

Data Requirements:

Reproducibility:

References

---

## [Decision Letter · Decision Letter 1]

16 Jun 2022

Dear Prof McCall,

We are pleased to inform you that your manuscript 'First comprehensive analysis of Aedes aegypti bionomics during an arbovirus outbreak in west Africa: dengue in Ouagadougou, Burkina Faso, 2016-2017' has been provisionally accepted for publication in PLOS Neglected Tropical Diseases.

Best regards,

Joseph T. Wu

Deputy Editor

Joseph Wu

Deputy Editor

Reviewer's Responses to Questions

**Key Review Criteria Required for Acceptance?**

**Methods**

-Are the objectives of the study clearly articulated with a clear testable hypothesis stated?

-Is the study design appropriate to address the stated objectives?

-Is the population clearly described and appropriate for the hypothesis being tested?

-Is the sample size sufficient to ensure adequate power to address the hypothesis being tested?

-Were correct statistical analysis used to support conclusions?

-Are there concerns about ethical or regulatory requirements being met?

Reviewer #1: (No Response)

Reviewer #2: (No Response)

**Results**

-Does the analysis presented match the analysis plan?

-Are the results clearly and completely presented?

-Are the figures (Tables, Images) of sufficient quality for clarity?

Reviewer #1: (No Response)

Reviewer #2: (No Response)

**Conclusions**

-Are the conclusions supported by the data presented?

-Are the limitations of analysis clearly described?

-Do the authors discuss how these data can be helpful to advance our understanding of the topic under study?

-Is public health relevance addressed?

Reviewer #1: (No Response)

Reviewer #2: (No Response)

**Editorial and Data Presentation Modifications?**

Reviewer #1: (No Response)

Reviewer #2: (No Response)

**Summary and General Comments**

Reviewer #1: (No Response)

Reviewer #2: In the first round of revisions I only had minor comments on the manuscript which have now all been addressed/corrected by the authors. Therefore I recommend publication of the paper.

PLOS authors have the option to publish the peer review history of their article (what does this mean?). If published, this will include your full peer review and any attached files.

Reviewer #1: **Yes: **Dr. Mang'era Clarence M.

Reviewer #2: No

---

## [Editor Report · Acceptance letter]

30 Jun 2022

Dear Prof McCall,

We are delighted to inform you that your manuscript, "First comprehensive analysis of *Aedes aegypti* bionomics during an arbovirus outbreak in west Africa: dengue in Ouagadougou, Burkina Faso, 2016-2017," has been formally accepted for publication in PLOS Neglected Tropical Diseases.

Best regards,

Shaden Kamhawi

co-Editor-in-Chief

Paul Brindley

co-Editor-in-Chief
